# Generation of large amplitude phonon states in quantum acoustics

Clinton A. Potts [1] ✉, Wilfred J. M. Franse[1], Victor Augusto S. V. Bittencourt [2], Anja Metelmann[2,3,4] & Gary A. Steele [1] ✉

The development of quantum acoustics has enabled the cooling of mechanical objects to their quantum ground state, generation of mechanical Fock-states, and Schrödinger cat states. Such demonstrations have made mechanical resonators attractive candidates for quantum information processing, metrology, and macroscopic tests of quantum mechanics. However, generating large-amplitude phonon states in quantum acoustic systems has been elusive. In this work, a single superconducting qubit coupled to a high-overtone bulk acoustic resonator is used to generate a large phonon population in an acoustic mode of a high-overtone resonator. We observe extended ringdowns of the qubit, confirming the generation of a large amplitude phonon state, and also observe an upper threshold behavior, a consequence of phonon quenching predicted by our model. This work provides a key tool for generating arbitrary phonon states in circuit quantum acoustodynamics, which is important for fundamental and quantum information applications.

The prospect of controlling and manipulating phonons at the quantum level has sparked considerable interest, in particular for applications of non-classical phonon states for quantum sensing[1–3]. The latter might enable tests of gravity effects in quantum mechanics, a long-pursued milestone in physics[4–6]. In this context, circuit quantum acoustodynamics (cQAD)[7] has emerged as a promising platform for controlling phonons at the quantum regime. In such systems, a single superconducting qubit is coupled to an acoustic phonon mode, enabling the use of the well-developed circuit quantum electrodynamics toolkit[8–10] for manipulating qubits as a way to control phonons[11–14]. As a consequence, cQAD has seen rapid success in the generation of mechanical states including the generation of Fock states[15] and Schrödinger cat states[16]. Moreover, cQAD has prospects for developing novel hybrid quantum systems[17–19], allowing further integration of the system with other hybrid architectures[20].

In this article, we demonstrate the generation of a large phonon population in cQAD using a scheme similar to single-atom lasers[21–23]. A single superconducting qubit couples to a bulk on-chip phonon mode,

generating a large amplitude phonon state through the process of stimulated emission. The phonon state amplitude is confirmed through a dramatically extended ringdown of the superconducting qubit excited by the phonon state. We also observe a unique predicted feature of our model, an upper threshold, which is a consequence of population quenching, a phenomenon also present in single-atom lasers[22,24,25]. Close to such an upper threshold, the phonon mode experiences a reduced linewidth, indicating a qubit-induced amplification. Our experimental observations agree with the theoretical and numerical models we use to characterize the system.

Finally, the phonon mode's bulk nature may allow the integration of color centers or quantum dots, enabling strain engineering or coherent mechanical driving for future hybrid quantum systems. Moreover, the coherent state generated here may be used as an efficient displacement pulse for generating large Schrödinger cat states[16] or squeezed states[26]. Generating large mechanical cat states has exciting potential applications for future macroscopic tests of quantum mechanics[27].

[1]Kavli Institute of Nanoscience, Delft University of Technology, PO Box 5046, 2600 GA Delft, The Netherlands. [2]Institut de Science et d'Ingénierie Supramoléculaires (ISIS, UMR7006), Université de Strasbourg and CNRS, 67000 Strasbourg, France. [3]Institute for Theory of Condensed Matter, Karlsruhe Institute of Technology, 76131 Karlsruhe, Germany. [4]Institute for Quantum Materials and Technology, Karlsruhe Institute of Technology, 76344 Eggenstein-Leopoldshafen, Germany. ✉e-mail: clinton.potts@nbi.ku.dk; g.a.steele@tudelft.nl

## Results

### Experimental setup

Our device comprises a flip-chip ℏBAR architecture with two bonded device chips[7,12,15,16,28,29]. We have implemented a fully on-chip integration, which can multiplex different devices on a single silicon chip[18,30]. A single feedline is coupled to the individual superconducting fixed-frequency transmon qubits. The transmon qubits were fabricated from niobium-titanium nitride for the bulk structures with aluminum Josephson junctions. A sapphire substrate, 650 $\mu$m thick, was positioned above each transmon qubit and bonded to the silicon substrate, see Fig. 1. The sapphire chip supports a set of longitudinal high-overtone bulk acoustic wave resonances (HBARs) separated by a free spectral range of 8.54 MHz. The electric field of a transmon qubit coherently couples to the strain of an HBAR acoustic mode via a disk of piezoelectric aluminum nitride patterned on the sapphire. Each qubit on the chip is nearly resonant with an HBAR mode of interest; in such a way, a pair of qubit-HBAR modes behaves like a single atom coupled to a phononic mode. The qubits were read out via on-chip microwave resonators using standard circuit quantum electrodynamics techniques[9,31]. Our device is similar to those used in previous work generating mechanical Schrödinger cat states[16] and for circuit quantum acoustodynamics[7].

Using the dispersive shift of a coupled linear readout resonator, we can measure the steady-state qubit population[9]. In the limit where the qubit and the readout resonator are far detuned in frequency, the qubit-cavity Hamiltonian can be written as:

$$\hat{\mathcal{H}}/\hbar = \omega_r \hat{a}^\dagger \hat{a} + \frac{1}{2}\omega_q \hat{\sigma}_z + \chi \hat{\sigma}_z \hat{a}^\dagger \hat{a}, \tag{1}$$

where $\omega_{r,q}$ are the readout and qubit frequency, $\hat{a}^{(\dagger)}$ is the photon annihilation (creation) operator, $\hat{\sigma}_z$ is the qubit Pauli-z operator, and $\chi$ is the qubit-state dependent frequency shift of the readout resonator. The coupling between the phonon and the qubit is described by a resonant Jaynes-Cummings interaction[7], given by the Hamiltonian:

$$\hat{\mathcal{H}}_{int}/\hbar = g_{qb}(\hat{\sigma}_+ \hat{b} + \hat{\sigma}_- \hat{b}^\dagger), \tag{2}$$

where $g_{qb}$ is the coupling rate between the qubit and the phonon mode, $\hat{\sigma}_\pm$ are the qubit raising and lowering operators, and $\hat{b}^{(\dagger)}$ is the phonon annihilation (creation) operator; see Fig. 2a.

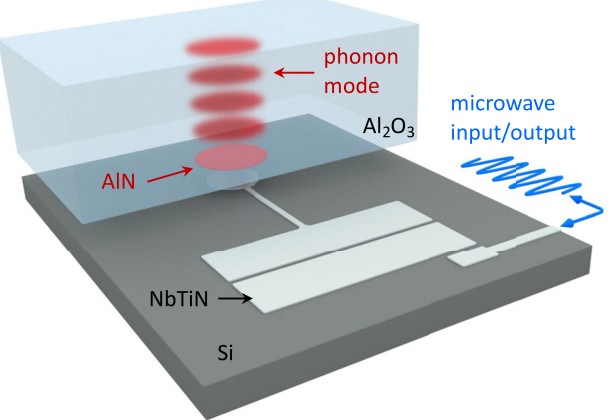

**Fig. 1 | Schematic of the on-chip ℏBAR device.** Rendering of the ℏBAR device. The ℏBAR device comprises two chips bonded in a flip-chip orientation. The top chip is 650 $\mu$m of sapphire, hosts the high-overtone bulk acoustic wave resonances (HBAR) modes (red), and is coupled to the superconducting antenna using an aluminum nitride pad (red). The pocket-style transmon qubit (silver) is fabricated from niobium titanium nitride on the bottom silicon chip and coupled to the feedline via an on-chip readout resonator (blue). Art produced by Enrique Sahagún[39].

The qubit's state was measured by applying a weak probe tone on resonance with the readout resonator and monitoring the transmitted signal as a second tone was swept near the qubit frequency. The transmission spectrum at the readout frequency directly maps to the qubit occupation $\langle \hat{\sigma}_+ \hat{\sigma}_- \rangle$; see Fig. 2b. The asymmetry of the qubit spectrum is due to the finite photon population within the readout resonator[32,33] and is well described by our theoretical model. Moreover, the narrow transparency window within the qubit spectrum results from the weak hybridization between the qubit and the phonon mode. In this work, the qubit and the phonon frequency were detuned by approximately 3.3 MHz. See the discussion in the supplementary text for full details. However, as will be discussed below, coherent energy exchange between the qubit and phonon modes does not provide a clear picture of the dynamics at high drive powers.

### Phonon Generation

Three distinct features can be observed within the two-tone spectrum when increasing the qubit drive power. First, the qubit linewidth is power-broadened[32,34]. Large qubit drive powers increase the qubit decay rate due to increased stimulated emission. Less intuitive is the gradual disappearance and narrowing of the phonon-induced transparency window; see Fig. 2c. The total linewidth of the transparency window is proportional to the phonon-qubit cooperativity and demonstrates an inverse dependence on the drive power; see Fig. 2d. Such behavior can not be entirely explained by the power broadening of the qubit linewidth. At powers closer to the minimum of the curve Fig. 2d, the narrowing of the transparency window linewidth can be associated with an amplification process triggered by the coupling to the driven qubit. Above such an upper threshold, the linewidth increases, a feature that can not be attributed to a suppression of the inverse Purcell effect due to the qubit broadening. These features can be understood by considering the schematic shown in Fig. 3a.

The build-up of the phonon population can be understood by considering the different processes through which excitations can be transferred between the phonon and the qubit. At low driving powers, the linewidth broadening of the qubit due to the microwave drive is small, and the phonon mode and qubit are weakly interacting. The weak interaction combined with the microwave drive initiates the buildup of the phonon population. A balance between the rate at which the qubit is excited and de-excited, the rate at which excitations are transferred between qubit and phonon mode and the stochastic phonon losses yields a steady-state occupation of the phonon mode. For low drive powers and thus slow qubit excitation rates, phonon loss dominates, preventing the build-up of the phonon population. As the qubit excitation rate increases, transitions between states of the Jaynes-Cummings ladder with higher phonon numbers can be efficiently driven. Since the rate at which excitations are transferred between states of the Jaynes-Cummings ladder depends on the phonon number of the involved states, as the average population of the phonon mode increases, the rate at which excitations are transferred correspondingly increases; this is the origin of acoustic stimulated emission[35].

In the steady state, this process, balanced by stochastic phonon losses, results in a large phonon population. Such processes also provide an intuitive understanding of the reduced visibility of the transparency window. As the drive power increases, the linewidth of the qubit becomes increasingly broad, and the coupling between the qubit and phonon is not strong enough to overcome the qubit loss; therefore, the qubit spectroscopy provides less information about the phonon mode.

At the highest qubit drive powers, the qubit linewidth has been increased such that the rapid decay of the qubit results in a quenching of phonon generation[24]. This results in an upper threshold above which the phonon mode is no longer effectively excited. Above this threshold, the qubit undergoes rapid Rabi oscillations due to the

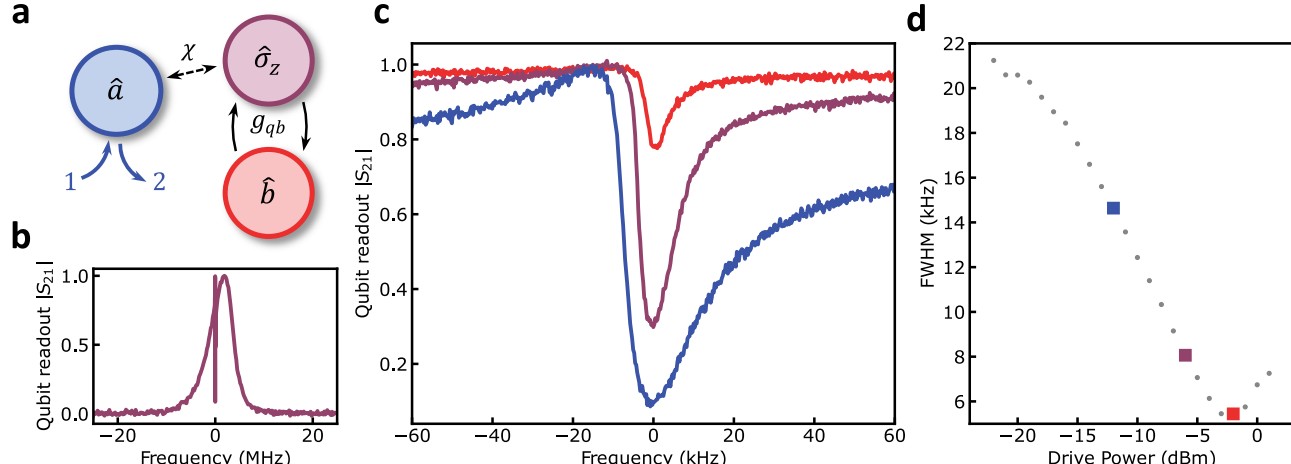

**Fig. 2 | Qubit-induced acoustic linewidth narrowing. a** Definition of modes and the coupling rates between the modes. The readout resonator $\hat{a}$ has input-output modes labelled 1 and 2, and is dispersively coupled to the qubit $\hat{\sigma}_z$ with a rate $\chi$. The mechanical mode is labeled $\hat{b}$ and has a qubit-phonon coupling rate $g_{qb}$. **b** Measured two-tone spectroscopy for drive power of −12.0 dBm set at room temperature. **c** Measured phonon-induced transparency window as a function of

qubit drive power. Starting from the lowest curve, drive powers are −12.0, −6.0, and −2.0 dBm set at room temperature. With increasing power, two features can be noticed. The transparency window reduces in depth, and the full-width half-maximum linewidth narrows. **d** Extracted experimental full-width half-maximum of the transparency window as a function of qubit drive power. Colored points match the corresponding curves in **c**.

strong microwave drive and, therefore, cannot exchange excitations with the phonon mode, reducing the phonon amplitude. The buildup of the phonon population, primarily due to stimulated emission from a single quantum emitter, including a reduced phonon mode linewidth and the upper threshold behavior, is similar to a single-atom laser. It should also be noted that the phonon mode statistics are no longer predicted to be that of a coherent state above the upper threshold. This process has been described previously in the context of single-atom photon lasers[21] and is captured by our theoretical description; see the supplementary text.

### Gated two-tone spectroscopy

Directly probing the phonon mode is not possible in the current experimental configuration since the readout is performed via the two-level system and not through a propagating photon mode[36,37]. Direct measurements of the Rabi oscillations between the qubit and phonon state have been previously used to measure Fock-states in $\hbar$BAR devices[15]. However, this data would not be possible due to the short lifetime of our qubit and the multiplicity of simultaneous Rabi oscillation frequencies between two Fock states is given by $2g_N = 2\sqrt{N}g_{qb}$, which scale with the phonon Fock number $N$.[15] Instead, we rely on the mismatch between the decay rate of the phonon mode $\tau \sim 25\,\mu s$ and that of the qubit. Using gated two-tone ringdown measurements, we can distinguish pure qubit decay from qubit decay driven by a highly excited phonon state. If the phonon mode is highly excited—in the absence of an external drive—the coherent Jaynes-Cummings interaction will continually drive the qubit, resulting in an extended relaxation of the qubit population compared to its intrinsic relaxation rate.

Gated two-tone measurements were performed, driving the qubit until the system reached its steady state; at this point, the drive was switched off using an RF switch. During the entire sequence, the frequency of the readout resonator is monitored using a vector network analyzer, averaging multiple traces triggered synchronously with the RF switch; see the supplementary text for more information. This measures the expectation value of the qubit population $\langle \hat{\sigma}_+ \hat{\sigma}_- \rangle$ as a function of time, with a temporal resolution of 50 ns.

First, the drive power was set near the peak of the lasing amplitude and was detuned 250 kHz above the HBAR resonance. The gated two-tone measurement was performed, and the blue data points in Fig. 3b show the resulting time domain measurement and a ringdown on the order of ~ 200 ns. This corresponds to the intrinsic $T_1$ decay of the

transmon qubit. A second measurement was performed at the same drive power, but the drive was tuned resonant with the HBAR. A ringdown on the order of 25 $\mu$s is observed for this configuration, represented by the red data points in Fig. 3b. The extended ringdown confirms the highly excited phonon amplitude of the mechanical state. When the qubit drive is switched off, the phonon mode can exchange excitations with the qubit, continually re-exciting the qubit until the phonon mode has decayed back to its ground state. Moreover, near the peak phonon amplitude, in contrast to a thermal state, the coherent state generated by the lasing results in a qubit ringdown that is not exponential; instead, the qubit ringdown is approximately linear. This feature is captured by our numerical model, the dashed curve in Fig. 3(b). Moreover, our model also captures the ring-up of the qubit, which is described in the supplementary information.

We can estimate the phonon population from our numerical model by fitting the spectroscopic and ringdown data. The estimated phonon population is shown in Fig. 4c as a function of qubit drive power. At a power of −2.0 dBm, the phonon population is nearly maximized, corresponding to the ringdown in Fig. 4a and b, and the upper threshold is clearly visible as the phonon population rapidly reduces with increasing drive power. The upper threshold is experimentally confirmed by performing a set of gated ringdown measurements at a series of qubit drive powers. With increasing qubit drive power, the individual ringdown traces grow in amplitude and increase in duration, corresponding to the increasing phonon population, as shown in Fig. 4a. At a drive power of −2.0 dBm, the phonon ringdown obtains its peak amplitude and duration, indicating a peak in the phonon population, shown in Fig. 4, which agrees with the minimum in the transparency window linewidth, shown in Fig. 2b. Further increasing the qubit drive power beyond −2.0 dBm, the qubit ringdown decreases in amplitude and duration, a direct indication of the self-quenching, well described by our theoretical model and similar to what has been observed in single-atom lasers, see Fig. 4b. The gated ringdown measurement demonstrates a clear upper threshold behavior and agrees with our numerical simulations and semi-classical analysis. We also notice that the qubit decay profile differs below and above the threshold for a given phonon population. Specifically, the decay is not exponential below the upper threshold, while well above the threshold, the decay is exponential. We associate such behavior with a change in the phonon state, which, according to our numerical simulations, is coherent below the upper threshold.

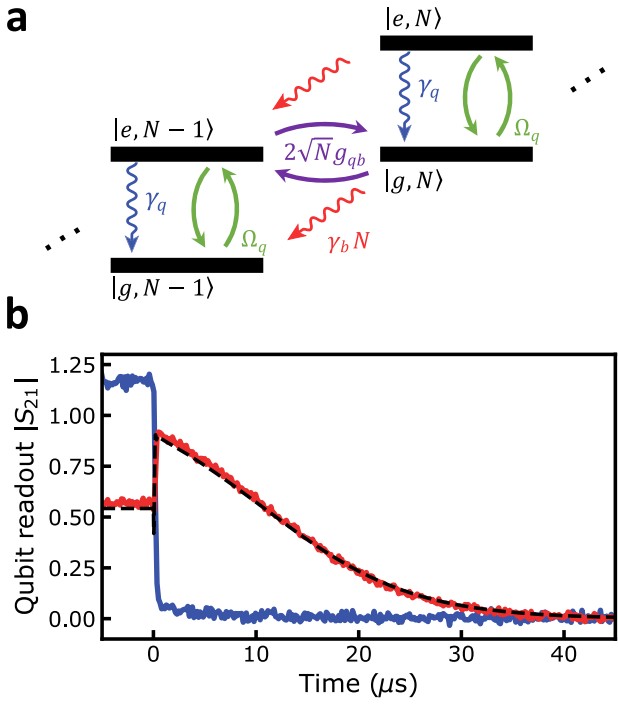

**Fig. 3 | Large amplitude phonon states observed through qubit ringdown dynamics. a** Schematic of the energy levels for the weakly hybridized phonon-qubit system. The qubit acts as an artificial two-level atom, with ground state $|g\rangle$ and excited state $|e\rangle$. A coherent drive of strength $\Omega_q$ drives the qubit between its ground and excited state. The qubit-mechanical couple at a rate $g_{qb}$ and a qubit and mechanics decay at a rate $\gamma_q$ and $\gamma_b$, respectively. The rate at which excitation transfer processes $|e, N-1\rangle \to |g, N\rangle$ occur scales as $\sqrt{N}$, where $N$ is the total number of excitations, while the phonon relaxation scales linearly with $N$. The qubit is rapidly re-excited for strong pump powers, resulting in a build-up of phonon excitations. **b** Measured qubit ringdown for a qubit drive of −3.0 dBm. Blue line: the qubit drive is detuned from the HBAR by 250 kHz with the phonon mode in the non-lasing state, decaying on a time scale of the -200 ns decay of the qubit. Red line: The qubit drive is tuned directly on resonance with the HBAR mode, exciting it into the lasing state, exhibiting a dramatically longer, non-exponential decay due to re-excitation from the coherently excited phonon mode. The master equation simulation (dashed black line) is plotted over the data.

## Discussion

This article demonstrates the experimental realization and generation of large-amplitude phonon states in cQAD. Our experiment consists of a superconducting single-atom, realized using a transmon-style qubit coupled resonantly to an HBAR mode. When driving on resonance with the HBAR mode, the intrinsic non-linearity of the qubit-phonon coupling generates a highly excited phonon state. A key feature of this experiment is the driven two-level atom rather than the parametric instability driving phonon excitations. Moreover, the phonon mode is confined in a bulk longitudinal mode within a sapphire substrate. The bulk nature may allow the phonon mode's integration with additional on-chip architectures, such as color centers or quantum dots.

Our results have demonstrated the successful generation of a large amplitude phonon state, and we have further demonstrated a counterintuitive feature predicted by our model, an upper threshold[25]. The size of the coherent state achieved in this work was limited by both the qubit and phonon linewidth. However, the primary limiting factor was the phonon linewidth. Decreasing the decay rates will reduce the upper threshold power and increase the peak phonon amplitude. Future studies could include a linear probe or use a higher-order transition of the transmon to drive and measure the phonon statistics. Such a cQAD-compatible system promises to provide a highly coherent source of phonons, which have applications including sensing to quantum information processing and the generation of high-displacement, Schrödinger cat states.

## Methods

### Device fabrication

**Qubit chip.** The device fabrication starts with a $10 \times 10$ mm chip $525\,\mu m$ thick high resistivity $\langle 100 \rangle$ silicon deposited with 100 nm of niobium-titanium nitride (NbTiN). The NbTiN film was deposited by the Dutch Institute for Space Research (SRON) following the process described in ref. 38. A layer of photoresist (AR-P 6200.18, 4000 rpm) was patterned, exposed (EBPG 5200, 315 $\mu m/cm^2$) and developed (Pentylacetate, O-xylene, IPA) to form the bulk circuitry (transmon islands and coplanar waveguides). The exposed NbTiN was removed using a reactive ion etch (Sentech Etchlab 200, 13.5 sccm $SF_6$ + 5 sccm $O_2$, 55 W, 10 $\mu$bar) followed by an in-situ oxygen descum (50 sccm $O_2$, 100 W, 10 $\mu$bar). After stripping the photoresist, a bilayer resist stack (MAA 8.5% EL6, 2000 rpm and PMMA A6 950k, 1500 rpm; baked for three and five minutes at 180 °C, respectively) was used for patterning the Josephson

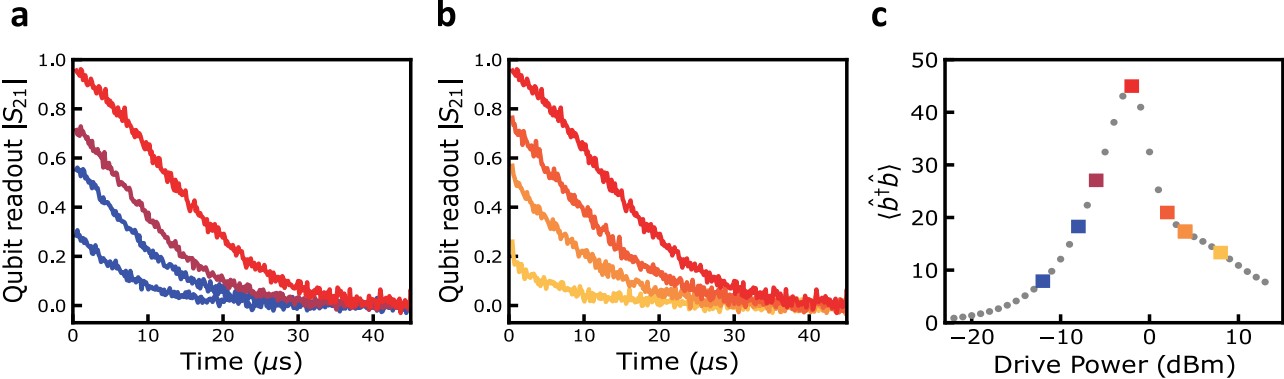

**Fig. 4 | Unique signatures of predicted upper threshold. a** Measured gated qubit ringdown for a qubit drive of −12.0, −8.0, −6.0, and −2.0 dBm. With increasing drive power, the gated qubit ringdowns increase in amplitude and duration as the phonon mode population increases. **b** Measured gated qubit ringdown for a qubit drive of −2.0, 2.0, 4.0, and 8.0 dBm. The qubit drive is tuned directly on resonance with the HBAR mode for all measurements. With increasing drive power, the gated qubit ringdowns decrease in amplitude and duration as the phonon mode population decreases above the self-quenching threshold. For all measurements in (**a**) and (**b**), the qubit drive is tuned directly on resonance with the HBAR mode. **c** Simulated phonon state population $\langle \hat{b}^\dagger \hat{b} \rangle$ as a function of qubit drive power. The phonon population is at its maximum at a drive power of approximately −2.0 dBm and decreases for drive powers above the upper threshold. The colored data points indicate the corresponding trace colors in (**a**) and (**b**).

junctions (190 nm width). The junctions were patterned using e-beam lithography. The bilayer was developed using cold $H_2O$ : IPA (1:3) and cleaned afterwards with IPA. After cleaning the exposed silicon surface with an oxygen descum (200 sccm, 100 W) and acid clean (BoE(7:1):$H_2O$, 1:1), the chip was placed in an aluminum evaporator (Plassys MEB550). Double-angle shadow evaporation with intermediate in-situ oxidation was used to create Manhattan-style junctions. The aluminum was evaporated at a 35° angle relative to the substrate at a rotational angle of 0° and 90°. The top and bottom electrodes were 35 and 75 nm thick, respectively. After the first evaporation step, the aluminum was oxidized to create the $AlO_x$ tunnel barriers. Following the second evaporation step, a second oxidation step was performed to cap the junctions with a passivation layer. After performing liftoff in NMP, the qubit chip was finished.

**HBAR chip.** The HBAR chip started with double-side polished four-inch sapphire wafers with a 1 $\mu$m thick film of c-axis oriented AlN (Kyma technologies, AT.U.100.1000.B). The wafer was diced into 10 × 10mm chips for easier processing. A photoresist layer (AR-N 4450.10, 6000 rpm) was used to pattern circular regions, 250$\mu$m in diameter, to mask the AlN. A reactive ion etch in an Oxford 100 was performed to create AlN disks ($Cl_2$/$BCl_3$/Ar at 4.0/26.0/10.0 sccm, 350 W ICP power, 70 W RF power). Following the reactive ion etch, the AlN layer has the proper shape but not the correct thickness. After stripping the photoresist, the chip was placed again inside the etcher to etch the AlN to ~ 900 nm thickness.

**Flip chip.** Once fabrication on both chips was done, the HBAR chip was diced into 8 × 2 mm chips. The HBAR chip was then flipped on top of the qubit chip with the AlN layer facing down. Using probe needles, the AlN disks were aligned with the transmon antennas. Once aligned, the probe needles held down the chips in position while a tapered fiber was used to apply two-component epoxy (Loctite EA 3430) on the sides of the top chip; see the supplementary information. After the epoxy was cured, the chip was wire-bonded and installed onto the baseplate of the dilution refrigerator.

### Measurement setup

**Two-tone spectroscopy.** All measurements were performed within a dilution refrigerator operating at a base temperature $T$ ~ 20 mK. A schematic of the dilution refrigerator setup and the room-temperature electronics are shown in the supplementary information. The device was mounted on the mixing chamber plate of the dilution refrigerator and connected to a set of coaxial cables. The device was measured in transmission, with the resonators coupled in a 'notch'-style geometry. The output signals went into a cryogenic HEMT (High Electron Mobility Transistor) amplifier (LNF-LNC4-8A), followed by additional room-temperature amplification (Miteq AFS3-04000800-07-10P-4). The input line was attenuated at each stage to reduce the electron temperature and the thermal radiation at the input port of our device. A total of 48 dB of attenuation was used, plus any additional attenuation from the coaxial cables.

The two-tone spectroscopy was measured using a vector network analyzer (VNA). Port one and port three were combined using a directional coupler, with port three attached to the -20 dB coupling port. Port one was set into zero span mode and output a constant signal tuned on resonance with the Stark shifted readout resonator, $\omega_r$, with an output power of -25 dBm. An additional 60 dB of attenuation was added to this signal before the directional coupler. Port 3 was used as a spectroscopic tone and was swept near the qubit frequency, and its power was varied throughout the experiment and had an additional 40 dB of attenuation. The combined signals from port one and there were then connected to the input line of the dilution refrigerator.

The output from the dilution refrigerator was directly connected to port 4 of the VNA set in zero span mode at the readout resonator

frequency $\omega_r$. Two-tone spectroscopy was performed by slowly sweeping the qubit drive tone, ensuring the system has reached its steady state and monitoring the readout resonators transmission spectrum $S_{21}$.

**Gated two-tone spectroscopy.** For the gated two-tone measurement, port one of the VNA was set up just as in the two-tone measurement. An external signal generator generated the qubit drive tone. The qubit drive tone was passed through an RF switch before being combined with the readout tone using a directional coupler. The RF switch was triggered using a pulse generator at a 3 kHz repetition rate. The signal generator was set to a 22.5% duty cycle, so the qubit drive was off for 75 $\mu$s per trace. The VNA was synchronously triggered by the signal generator, allowing for 65536 trace averages to be performed (maximum setting). Each trace-averaged measurement was repeated 75 additional times to improve the signal-to-noise ratio further.

## Data availability
All data, analysis code, and measurement software are available in the manuscript or the supplementary material or are available at Zenodo https://doi.org/10.5281/zenodo.14810526.

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

## Acknowledgements

The authors thank Enrique Sahagun for the device rendering[39]. W.J.M.F and G.A.S. acknowledge support through the QUAKE project, project number 680.92.18.04, of the research programme Natuurkunde Vrije Programma's of the Dutch Research Council (NWO). C.A.P. acknowledges the support of the Natural Sciences and Engineering Research Council of Canada (NSERC) (PDF-567689-2022). A.M and V.A.S.V.B. acknowledge financial support from the Contrat Triennal 2021-2023 Strasbourg Capitale Europeenne.

## Author contributions

C.A.P. performed experiments, theoretical modelling, conceptualization, and wrote the manuscript with input from all authors. W.J.M.F. fabricated the device and performed experiments. V.A.S.V.B. performed theoretical modelling. A.M. provided supervision and funding acquisition. G.A.S. provided supervision, conceptualization and funding acquisition.

## Competing interests

The authors declare no competing interests.

## Additional information

**Peer review information** : *Nature Communications* thanks the anonymous reviewers for their contribution to the peer review of this work. A peer review file is available.

