## [Transparent Peer Review file · Nature Communications]

Generation of Large Amplitude Phonon States in Quantum Acoustics

Corresponding Author: Dr Clinton Potts

Version 0:

Reviewer comments:

Reviewer #1

(Remarks to the Author)

In the manuscript "A Superconducting Single-Atom Phonon Laser", the authors report the observation of large amplitude driving of a HBAR mode, enabled by its coupling to a superconducting qubit. They explore the dynamics of this system, which show interesting behavior such as the initial increase of phonon number with drive power, followed by a decrease. They also show that the excitations in the system decay with the lifetime of the HBAR mode instead of the qubit, which is evidence that they are indeed observing phonon populations. They claim that these observations, among others, are signatures of a single-atom phonon laser, which can have applications in, for example, the preparation of Schrödinger cat states.

The experimental techniques are well-explained and seem sound, and the data is clearly presented. I think that, overall, these results are interesting for the cQAD community, but I find that some of the claims are not so well supported.

First, the central claim of the manuscript is that this device behaves as a laser, and the authors give a description of the physical process, involving the transfer of excitations between the qubit and the HBAR mode, that invoke laser-related phenomena such as stimulated emission. However, the authors also say that at low driving powers, the phonon mode and qubit hybridize weakly. First of all, I don't quite understand why the hybridization of two modes depend on the driving power. Are they implying that the hybridization increases at higher power? In the standard cavity QED model, doesn't the hybridization just depend on the coupling strength and detuning? Second, given that the phonon mode hybridizes with the qubit, can't it just be directly driven by the microwave drive? This would also lead to an increase in the steady state phonon population with drive power. The linewidth narrowing can also be explained by the effective decay rate of the phonons through the qubit. In the regime that the two are detuned by less than the qubit linewidth, this effective rate scales inversely with the qubit linewidth, so that it decreases as the qubit is power broadened, until some other loss rate becomes dominant. These other loss mechanisms could also increase at very large drive power due to, say, heating, resulting in a broadening of the mechanical linewidth and decrease in the phonon population. I wouldn't call any of these effects lasing.

Another claim the authors make is that a coherent state is created in the HBAR mode, but they don't show any phonon statistics or state tomography measurements to support this. In the supplementary information, they show simulations that indicate that a coherent state with a well-defined phase is indeed created. However, it's far from clear that these simulations capture exactly what is going on in the experiment, including all sources of noise. The claim of making a coherent state is a bit weak without experimental data.

To summarize, I think the results of this work are interesting and should be published eventually. However, to me, it's not conclusive that they can only be interpreted by invoking the concepts of lasing and generation of coherent states. As for the usefulness for, say, quantum state generation, this of course depends on whether the authors can show conclusive experimental evidence of a large coherent state, but I don't think it matters if one calls it a laser.

One minor comment: The figure captions in the supplementary information refer to panels as (A), (B), etc., but there are no labels in the figures.

Reviewer #2

(Remarks to the Author)

The manuscript "A superconducting Single-Atom Phonon Laser" claims to experimentally demonstrate single-atom lasing behaviour in a circuit quantum acoustodynamical system where the single atom is mimicked by a superconducting qubit and the role of the lasing field is played by a mode of bulk acoustic wave resonances within a sapphire chip coupled through a piezoelectric cladding of the chip.

Firstly, the authors show that the phonon-induced transparency window measured through two-tone spectroscopy becomes shallower and narrower with increasing drive power. It is then shown that the phonon mode is populated due to excitation transfer from the driven qubit into the phonon mode, balanced by phonon decay. This transfer is deemed as the stimulated emission process and serves as the first signature of, or evidence for, single-atom lasing behaviour in this system.

Further increasing the drive power leads to an increase in the qubit linewidth causing a decrease in the transfer of excitations into the phonon mode. This yields an upper threshold in power for the lasing state, which is another signature of single-atom lasing.

The first main technical issue of this manuscript lies in the (absence of the) third and final signature of single-atom lasing, which would lie in the intensity correlations of the lasing field. There is a theoretical discussion at the end of Sec. II of the supplemental material, together with the theoretical simulation shown in Fig. S7(c), indicating Poissonian statistics. This is interpreted as the creation of coherent states in the phonon mode. The manuscript displays well-executed experimental and theoretical work which, however, makes the claim that they "we experimentally demonstrate a direct quantum-acoustic equivalent of a single-atom laser," without showing the evidence of sub-Poissonian statistics in the intensity correlation of the lasing field. By comparison, examples of single-atom lasing typically demonstrate this effect; cf. [Nature 425, 268 (2003)], which shows sub-Poissonian intensity correlations in the for single-atom lasing regime, and [Nature Phys. 6, 350 (2010)], which shows an increase of the intra-cavity mean excitation number in combination with decreasing intensity fluctuations in the sub-Poissonian regime and calls these measurements "the signature of a thresholdless single-atom laser." At a minimum, the present manuscript is missing a thorough discussion of why, despite the claim of demonstrating the "direct quantum-acoustic equivalent of a single-atom laser," equivalent intensity correlations are not presented; this should be combined with a discussion on the experimental methods that may be employed to measure such intensity correlations. That being said, actual experimental measurement of the intensity would be more desirable. Additionally, a short outline of this discussion should be part of the main manuscript, together with longer discussion in the supplemental material, for the benefit of the general audience like the one Nature Communications reaches.

The second main technical issue of this manuscript regards the phase seeding, which is discussed in Sec. III B of the Supplementary Information. For a lasing system described by a parametric instability, the phase typically cannot be seeded by the drive; and to the best of my knowledge the same holds for single-atom lasers. The authors should discuss clearly why this is not the case in the present scenario and point to other examples in the literature, if available, where this phase seeding by the driving field was present.

Otherwise, the manuscript is clearly organised and well-written in general, with a few minor exceptions:

1. There are some inconsistencies with the documentation of the frequencies and rates; the manuscript and supplementary information do not always have the factor of 2π explicitly indicated. The authors are encouraged to use a consistent convention throughout in order to ensure reproducibility; the current state of the manuscript may make it hard to reproduce the data.

2. Supplementary Figures S5, S6, S7, S8, and S9 refer to subfigures enumerated with letters without these being indicating in the figure itself, which makes reading and interpreting the captions unnecessarily complicated.

To conclude, I cannot recommend this manuscript for publication in Nature Communications in its current state unless these concerns regarding the intensity correlation statistics, the phase seeding, and the minor issues are answered and changes implemented in a convincing fashion.

Reviewer #4

(Remarks to the Author)

Version 1:

Reviewer comments:

Reviewer #1

(Remarks to the Author)

The authors have addressed my concerns from the previous round of reviews. Given the re-worded claims and title, I would

now support publication.

Reviewer #2

(Remarks to the Author)

In the revised version of the manuscript "Large Amplitude Phonon States in Quantum Acoustics", the authors have made great improvements regarding the clarity of the results by backing down from claiming the observation of single-atom lasing behaviour. I agree that the generation of large amplitude phonon states in cQAD is an advance in the field and carries potential for non-Gaussian state creation. However I disagree that this constitutes enough of an advance to be published in Nature Communications.

In summary, I cannot recommend this manuscript for publication in Nature Communications. Nevertheless, I want to congratulate the authors on their manuscript and recommend it for publication in Scientific Reports in its current state after a revision of a few minor unclarities outlined below.

Minor comments on the manuscript:

1. Supplementary Figure S10 still has the caption title "Phonon Lasing Phase Seeding" even though the authors do not claim that any lasing is taking place.
2. The labelling of the subfigures in the Supplemental material is still inconsistent. Fig. S2 carries labels in capitals while the caption refers to the subfigures in small letters; Fig. S9 consists of one figure which has no sublabel while the caption refers to a subfigure (A). A streamlined labelling convention throughout the entire document would improve its legibility.
3. The editorial policy checklist file has checked the box "We declare that one or more of the authors have a competing interest as defined by Nature Portfolio" while the manuscript says in the Ethics declarations "The authors declare no competing interests". This seems contradictory to me and requires an elaboration.

Reviewer #4

(Remarks to the Author)

Referee Comments
Nature Communications

We hereby resubmit our article and a response to the referees' comments. The referee reports were mainly concerned with the lack of direct experimental evidence supporting the claim that our system is a single-atom laser. In particular, it was not possible to access the phonon state statistics in the experiment, which weakens our claim that the phonon state is coherent below the threshold. While our system still exhibits different traits of a single-atom laser, including the upper threshold behavior and the linewidth reduction below its intrinsic value, we agree with the referees that further experimental evidence would support our claim. We would like, nevertheless, to point out that both referees evaluated our work positively, with referee 1 stating that our work is valuable for the cQAD community.

Following the referee's comments, we have decided to remove the strong language regarding a single-atom phonon laser and give the article a new title, "Generation of Large Amplitude Phonon States in Quantum Acoustics." We are grateful for the thoughtful referee comments and believe that by answering these questions, we have substantially improved the quality of our article.

Please see our replies to the referees' questions below. All changes in the main text are highlighted in blue, and our replies to the referees in this letter are likewise highlighted in blue.

Reviewer: 1

In the manuscript "A Superconducting Single-Atom Phonon Laser", the authors report the observation of large amplitude driving of a HBAR mode, enabled by its coupling to a superconducting qubit. They explore the dynamics of this system, which show interesting behavior such as the initial increase of phonon number with drive power, followed by a decrease. They also show that the excitations in the system decay with the lifetime of the HBAR mode instead of the qubit, which is evidence that they are indeed observing phonon populations. They claim that these observations, among others, are signatures of a single-atom phonon laser, which can have applications in, for example, the preparation of Schrödinger cat states.

The experimental techniques are well-explained and seem sound, and the data is clearly presented. I think that, overall, these results are interesting for the cQAD community, but I find that some of the claims are not so well supported.

We thank the referee for their positive evaluation of the manuscript and of the results, in particular the importance of our results in the context of cQAD. We agree that our previous claims would be strengthened with more experimental evidence, which was unfortunately inaccessible. We have amended the text and focused on the fact that in our experiment we have observed signatures of the generation of a large phonon population, as well as the upper threshold behavior of the system, in conjunction with excellent agreement between our theoretical model and experi-

mental data.

First, the central claim of the manuscript is that this device behaves as a laser, and the authors give a description of the physical process, involving the transfer of excitations between the qubit and the HBAR mode, that invoke laser-related phenomena such as stimulated emission. However, the authors also say that at low driving powers, the phonon mode and qubit hybridize weakly. First of all, I don't quite understand why the hybridization of two modes depend on the driving power. Are they implying that the hybridization increases at higher power? In the standard cavity QED model, doesn't the hybridization just depend on the coupling strength and detuning? Second, given that the phonon mode hybridizes with the qubit, can't it just be directly driven by the microwave drive? This would also lead to an increase in the steady state phonon population with drive power. The linewidth narrowing can also be explained by the effective decay rate of the phonons through the qubit. In the regime that the two are detuned by less than the qubit linewidth, this effective rate scales inversely with the qubit linewidth, so that it decreases as the qubit is power broadened, until some other loss rate becomes dominant. These other loss mechanisms could also increase at very large drive power due to, say, heating, resulting in a broadening of the mechanical linewidth and decrease in the phonon population. I wouldn't call any of these effects lasing.

We thank the referee for the technical questions. First, we should point out that, in our system, the qubit and the phonon mode are *not hybridized*. The parameters of the system are such that the qubit relaxation rate Γ_1 is larger than the qubit-phonon coupling g_{qb} , placing our experiment in the weak-coupling regime. Such a fact is also prominent in the qubit spectrum shown in the SI figure S7: there is no indication of mode-splitting, which would be a hallmark of hybridization. We have changed this information through the text to "weakly interacting" instead of "weakly hybridized". Another remark we would like to make is that the qubit experiences a linewidth broadening with an increase in drive power. This reduces the qubit-phonon *cooperativity* at increased drive powers.

Indeed, part of the linewidth narrowing comes from the effective decay rate of the phonons through the qubit, an inverse Purcell effect, which is suppressed as the power is increased, and the qubit is power broadened. However, this mechanism alone can not explain that the linewidth narrowing goes *below* the intrinsic phonon linewidth until the upper threshold is reached, then increases above the upper threshold.

If we consider the model we discuss in the SI of the manuscript and calculate the total linewidth of the transparency window, we observe the linewidth drops below the intrinsic linewidth. Mean field calculations of the system dynamics also support such an observation. These calculations consist of solving numerically the mean-field dynamics of the system, as given by Eqs. S11, which gives us the time evolution of the phonon amplitude. We then Fourier transform the results to calculate the phonon spectrum and extract the linewidth by fitting a Lorentzian function. We note the linewidth narrowing below the intrinsic phonon linewidth, for powers close to the upper threshold, as shown in the Supplementary Figure SI 5, which we reproduce here in Fig. 1. We also observe in the experimental data, from our time-domain fit, that the total linewidth of the transparency window at the upper threshold is less than the intrinsic linewidth of the phonon mode, agreeing with our mean-field model. This provides experimental evidence that the linewidth narrowing mechanism is not

simply a suppression of the inverse Purcell effect, and is in fact related to non-linearities stemming from the Jaynes-Cummings interaction.

Invoking spurious heating of the phonon mode by other sources to explain the upper threshold behavior is unnecessary, since our model predicts such a behavior and, furthermore, the upper threshold power corresponds exactly with the experimental observation in the qubit ringdown experiment. In fact, the maximum phonon population predicted by our model occurs for a power at which we *experimentally observed* the longest qubit ringdown, which is also the power at which the transparency window exhibits the minimum linewidth. The measurement of the transparency window linewidth is *completely independent* from the qubit ringdown measurement. We would like to emphasize that our minimal model explains all the observed behavior of the data.

Another claim the authors make is that a coherent state is created in the HBAR mode, but they don't show any phonon statistics or state tomography measurements to support this. In the supplementary information, they show simulations that indicate that a coherent state with a well-defined phase is indeed created. However, it's far from clear that these simulations capture exactly what is going on in the experiment, including all sources of noise. The claim of making a coherent state is a bit weak without experimental data.

We agree with the referee that our claim is weakened without direct evidence of a phononic coherent state. Unfortunately, obtaining any second-order correlation or

Figure 1: **Linewidth narrowing of the HBAR mode** (A) Steady-state phonon occupation for the system in the manuscript. (B) Effective phonon linewidth extracted from a Lorentzian fit of the phonon spectrum. (C,D) phonon spectra at two representative drive amplitudes showing the linewidth narrowing with power, the black dashed line depicts the spectrum for a phonon mode uncoupled from the two-level system. Parameters in correspondence with the experiment.

performing phonon tomography in our experimental setup was not possible. Nevertheless, we would like to point out that the different qubit decay profiles above and below the upper threshold are related to the phonon state and provide indirect evidence that the phonon steady-state is different below and above the threshold. We have amended the text and toned down our claims.

To summarize, I think the results of this work are interesting and should be published eventually. However, to me, it's not conclusive that they can only be interpreted by invoking the concepts of lasing and generation of coherent states. As for the usefulness for, say, quantum state generation, this of course depends on whether the authors can show conclusive experimental evidence of a large coherent state, but I don't think it matters if one calls it a laser.

We thank the referee for their helpful comments and suggestions. We have removed the claim that we have created a laser. Indeed, we would like to have tomography of the phonon mode; however, in our current experiment, this was not possible. Indeed, to perform tomography, one must be able to perform large displacements of the phonon state, which, before this work, was not possible. Yet we believe we have proposed a concrete and simple model to describe our experimental setup. We can describe all observed results from this most simplistic model in this manuscript. This suggests that we have generated large coherent phonon states and observed interesting dynamics of an upper threshold for phonon generation.

One minor comment: The figure captions in the supplementary information refer to panels as (A), (B), etc., but there are no labels in the figures.

Thank you for pointing this out. We have updated the figures accordingly.

Reviewer: 2

The manuscript "A superconducting Single-Atom Phonon Laser" claims to experimentally demonstrate single-atom lasing behaviour in a circuit quantum acoustodynamical system where the single atom is mimicked by a superconducting qubit and the role of the lasing field is played by a mode of bulk acoustic wave resonances within a sapphire chip coupled through a piezoelectric cladding of the chip.

Firstly, the authors show that the phonon-induced transparency window measured through two-tone spectroscopy becomes shallower and narrower with increasing drive power. It is then shown that the phonon mode is populated due to excitation transfer from the driven qubit into the phonon mode, balanced by phonon decay. This transfer is deemed as the stimulated emission process and serves as the first signature of, or evidence for, single-atom lasing behaviour in this system.

Further increasing the drive power leads to an increase in the qubit linewidth causing a decrease in the transfer of excitations into the phonon mode. This yields an upper threshold in power for the lasing state, which is another signature of single-atom lasing.

We thank the referee for their evaluation of our manuscript and their helpful comments.

The first main technical issue of this manuscript lies in the (absence of the) third and final signature of single-atom lasing, which would lie in the intensity correlations of the lasing field. There is a theoretical discussion at the end of Sec. II of

the supplemental material, together with the theoretical simulation shown in Fig. S7(c), indicating Poissonian statistics. This is interpreted as the creation of coherent states in the phonon mode. The manuscript displays well-executed experimental and theoretical work which, however, makes the claim that they “we experimentally demonstrate a direct quantum-acoustic equivalent of a single-atom laser,” without showing the evidence of sub-Poissonian statistics in the intensity correlation of the lasing field. By comparison, examples of single-atom lasing typically demonstrate this effect; cf. [Nature 425, 268 (2003)], which shows sub-Poissonian intensity correlations in the for single-atom lasing regime, and [Nature Phys. 6, 350 (2010)], which shows an increase of the intra-cavity mean excitation number in combination with decreasing intensity fluctuations in the sub-Poissonian regime and calls these measurements “the signature of a thresholdless single-atom laser.” At a minimum, the present manuscript is missing a thorough discussion of why, despite the claim of demonstrating the “direct quantum-acoustic equivalent of a single-atom laser,” equivalent intensity correlations are not presented; this should be combined with a discussion on the experimental methods that may be employed to measure such intensity correlations. That being said, actual experimental measurement of the intensity would be more desirable. Additionally, a short outline of this discussion should be part of the main manuscript, together with longer discussion in the supplemental material, for the benefit of the general audience like the one Nature Communications reaches.

We agree that the absence of phonon statistics is unfortunate and would have strengthened our claims. However, unlike the two articles presented, the phonon mode is confined within the bulk substrate and does not propagate into free space, allowing for intensity correlations to be measured. Unfortunately, the only probe of the phonons is the qubit we drive to generate the large phonon state.

In the experiment, it was not possible to directly access the phonon mode, and the setup was not suitable for implementing a qubit-assisted tomography of the phonon state. Nevertheless, the qubit decay profile gives indirect evidence that the statistics of the phonon steady-state below the upper threshold are different than those above the upper threshold. Unfortunately, it is also not trivial to directly relate the qubit decay profile to, e.g. the phonon intensity correlation, since our device works in a regime in which it is not possible to adopt simplifications like adiabatic elimination of one of the modes. We have added comments to the text about this point.

In light of both referees’ comments, we have decided to remove the phonon lasing language as this has been a strongly contested claim. We are confident we have generated a large phonon state, which is indeed due to amplification resulting from the coupled two-level system. However, experimentally, we cannot measure the many properties of a single-atom laser.

The second main technical issue of this manuscript regards the phase seeding, which is discussed in Sec. III B of the Supplementary Information. For a lasing system described by a parametric instability, the phase typically cannot be seeded by the drive; and to the best of my knowledge the same holds for single-atom lasers. The authors should discuss clearly why this is not the case in the present scenario and point to other examples in the literature, if available, where this phase seeding by the driving field was present.

We would like to point out that the generation of the large amplitude phonon state

here is not a result of a parametric instability. We agree that a drive does not set the phase of a parametric instability. Parametric instability measured in the IQ plane generates a ring of constant amplitude with no defined phase. Instead, the large amplitude here is due to stimulated emission of phonons from the superconducting qubit, which has a well-defined phase. Again, we have removed the language of single-atom lasing, as what we have described in the model indicates that the drive's phase seeds the phonon state's phase. This property is desired as a well-defined phase of the phonon state is desired, for example, to generate a cat state.

Otherwise, the manuscript is clearly organised and well-written in general, with a few minor exceptions:

1. There are some inconsistencies with the documentation of the frequencies and rates; the manuscript and supplementary information do not always have the factor of 2π explicitly indicated. The authors are encouraged to use a consistent convention throughout in order to ensure reproducibility; the current state of the manuscript may make it hard to reproduce the data.

We thank the referee for noticing these oversights, and this has been corrected in the resubmitted version of the manuscript

2. Supplementary Figures S5, S6, S7, S8, and S9 refer to subfigures enumerated with letters without these being indicating in the figure itself, which makes reading and interpreting the captions unnecessarily complicated.

This figures have been updated to contain the proper enumerations

To conclude, I cannot recommend this manuscript for publication in Nature Communications in its current state unless these concerns regarding the intensity correlation statistics, the phase seeding, and the minor issues are answered and changes implemented in a convincing fashion.

Reviewer: 3

We would like to thank the referees again for their comments and corrections. We have amended the text and toned down our previous claims. We hope that our answer has clarified some conceptual points and that the new version of the manuscript can be recommended for publication.

Sincerely,

Dr. Clinton Potts
NQCP Post-Doctoral Researcher, University of Copenhagen
clinton.potts@nbi.ku.dk

Prof. Dr. Gary A. Steele
Antoni van Leeuwenhoek Professor
Department of Quantum Nanoscience, Delft University of Technology
g.a.steele@tudelft.nl

Referee Comments
Nature Communications

We hereby resubmit our article and a response to the referees' comments. We thank the referees for taking the time to consider our work and provide helpful scientific comments that helped improve the quality of our article.

Reviewer: 1

The authors have addressed my concerns from the previous round of reviews. Given the re-worded claims and title, I would now support publication.

We thank the referee for their support for the publication of our article.

Reviewer: 2

In the revised version of the manuscript "Large Amplitude Phonon States in Quantum Acoustics", the authors have made great improvements regarding the clarity of the results by backing down from claiming the observation of single-atom lasing behaviour. I agree that the generation of large amplitude phonon states in cQAD is an advance in the field and carries potential for non-Gaussian state creation. However I disagree that this constitutes enough of an advance to be published in Nature Communications.

In summary, I cannot recommend this manuscript for publication in Nature Communications. Nevertheless, I want to congratulate the authors on their manuscript and recommend it for publication in Scientific Reports in its current state after a revision of a few minor unclarities outlined below.

We thank the referee for their comments that helped improve the clarity of our article. However, we respectfully disagree; we believe this work demonstrates a non-trivial feature of cQAD to date that has not been shown previously in the literature. Providing a tool which has previously been described in literature as required for the generation of large non-Gaussian states. Therefore, we believe this article is suitable for the readership of Nature Communications, and agree with the opinion of Referee 1.

Minor comments on the manuscript: 1. Supplementary Figure S10 still has the caption title "Phonon Lasing Phase Seeding" even though the authors do not claim that any lasing is taking place.

We thank the referee for their careful reading of our manuscript and have updated the SI appropriately to remove all lasing language.

2. The labelling of the subfigures in the Supplemental material is still inconsistent.

Fig. S2 carries labels in capitals while the caption refers to the subfigures in small letters; Fig. S9 consists of one figure which has no sublabel while the caption refers to a subfigure (A). A streamlined labelling convention throughout the entire document would improve its legibility.

We have gone through the SI and updated all captions to streamline the notation.

3. The editorial policy checklist file has checked the box "We declare that one or more of the authors have a competing interest as defined by Nature Portfolio" while the manuscript says in the Ethics declarations "The authors declare no competing interests". This seems contradictory to me and requires an elaboration.

This was a mistake on our part in filling out the checklist. This has been updated on the final version of the Nature checklist as we declare no competing interests.

Reviewer: 3

We would like to thank the referees again for their comments and corrections. We have amended the text and toned down our previous claims. We hope that our answer has clarified some conceptual points and that the new version of the manuscript can be recommended for publication.

Sincerely,

Dr. Clinton Potts
NQCP Post-Doctoral Researcher, University of Copenhagen
clinton.potts@nbi.ku.dk

Prof. Dr. Gary A. Steele
Antoni van Leeuwenhoek Professor
Department of Quantum Nanoscience, Delft University of Technology
g.a.steele@tudelft.nl